# EGRU: Event-based GRU
# for activity-sparse inference and learning

## Abstract

The scalability of recurrent neural networks (RNNs) is hindered by the sequential dependence of each time step's computation on the previous time step's output. Therefore, one way to speed up and scale RNNs is to reduce the computation required at each time step independent of model size and task. In this paper, we propose a model that reformulates Gated Recurrent Units (GRU) as an event-based activity-sparse model that we call the Event-based GRU (EGRU), where units compute updates only on receipt of input events (event-based) from other units. When combined with having only a small fraction of the units active at a time (activity-sparse), this model has the potential to be vastly more compute efficient than current RNNs. Notably, activity-sparsity in our model also translates into sparse parameter updates during gradient descent, extending this compute efficiency to the training phase. We show that the EGRU demonstrates competitive performance compared to state-of-the-art recurrent network models in real-world tasks, including language modeling while maintaining high activity sparsity naturally during inference and training. This sets the stage for the next generation of recurrent networks that are scalable and more suitable for novel neuromorphic hardware.

## 1 Introduction

Large scale models such as GPT-3 [8], switch transformers [17] and DALL-E [52] have demonstrated that scaling up deep learning models to billions of parameters cannot just improve the performance of these models but lead to entirely new forms of generalisation. For example, GPT-3 can do basic translation and addition even though it was trained only on next word prediction. While it is unknown if scaling up recurrent neural networks can lead to similar forms of generalisation, the limitations on scaling them up preclude studying this possibility. The dependence of each time step's computation on the previous time step's output is the source of a significant computational bottleneck, preventing RNNs from scaling well. Therefore, in recent years, RNNs, despite their many desirable theoretical properties [15] such as the ability to process much longer context and their computational power [57, 60], have been supplanted by feedforward network architectures.

By reducing the computation required at each time step, independent of model size and task, we can speed up and better scale RNNs. We propose to do this by designing a general-purpose event-based recurrent network architecture that is naturally activity-sparse. Dubbed the Event-based Gated Recurrent Unit (EGRU), our model is an extension of the Gated Recurrent Unit (GRU) [12]. With event-based communication, units in the model can decide when to send updates to other units, which then trigger the update of receiving units. Therefore, network updates are only performed at specific, dynamically determined event times. With activity-sparsity, most units do not send updates to other units most of the time, leading to substantial computational savings during training and inference. We formulate the gradient updates of the network to be sparse using a novel method, extending the benefit of the computational savings to training time.

The biological brain, which relies heavily on recurrent architectures and is at the same time extremely energy efficient [43], is a major source of inspiration for the EGRU. One of the brain's strategies to reach these high levels of efficiency is activity-sparsity. In the brain, (asynchronous) event-based communication is just the result of the properties of the specific physical and biological substrate on which the brain is built. Biologically realistic spiking neural networks and neuromorphic hardware also aim to use these principles to build energy-efficient software and hardware models [53, 58]. However, despite progress in recent years, their task performance has been relatively limited for real-world tasks compared to state-of-the-art recurrent architectures based on LSTM and GRU. We view the EGRU as a generalisation of spiking neural networks, moving away from modeling biological dynamics toward a more general-purpose recurrent model for deep learning.

In this paper, we first introduce a version of EGRU based on a principled mathematical approach that formulates the dynamics of the internal states of the network in continuous time. The units of the network communicate solely through message events triggered when the internal state of a unit reaches a threshold value. This allows us to derive exact gradient descent update equations for the network analogous to backpropagation-through-time (BPTT) that mirrors the activity-sparsity of the forward pass.

We then introduce a discrete simplification of this continuous-time model that is also event-based and activity-sparse while being easier to implement on today's prevailing machine learning libraries and thus directly comparable to existing implementations of GRU and LSTM. The backwards pass here uses an approximate version of BPTT, and these updates are also sparse.

The sparsity of the backward-pass overcomes one of the major roadblocks in using large recurrent models, which is having enough computational resources to train them. We demonstrate the task performance and activity sparsity of the model implemented in PyTorch, but this formulation will also allow the model to run efficiently on off-the-shelf hardware, including CPU-based nodes when implemented using appropriate software paradigms. Moreover, an implementation on novel neuromorphic hardware like [13, 27], that is geared towards event-based computation, can make the model orders of magnitude more energy efficient [48].

In summary, the main contributions of this paper are the following:

1. We introduce the EGRU, an event-based continuous-time variant of the GRU model.

2. We derive an event-based form of the error-back-propagation algorithm for EGRU.

3. We introduce a discrete-time version of EGRU that can be directly compared to current LSTM/GRU implementations.

4. We demonstrate that the EGRU exhibits task-performance competitive with state-of-the-art recurrent network architectures (based on LSTM, GRU) on real-world machine learning benchmarks.

5. We show that EGRU exhibits high levels of activity-sparsity during both inference and learning.

## 2   Related work

Activity sparsity in RNNs has been proposed previously in various forms [28, 46, 47], but only focusing on achieving it during inference. Conditional computation is a form of activity sparsity used in [17] to scale to 1 trillion parameters. This architecture is based on the feedforward transformer architecture, with a separate network making the decision of which sub-networks should be active [59]. An asynchronous event-based architecture was recently proposed specifically targeted towards graph neural networks [56]. QRNNs [7], SRUs[38] and IndRNNs [39] target increasing the parallelism in a recurrent network without directly using activity-sparsity. Unlike [17], our architecture uses a unit-local decision making process for the dynamic activity-sparsity, specifically for recurrent architecture. The cost of computation is lower in an EGRU compared to [47], and can be implemented to have parallel computation of intermediate updates between events, while also being activity sparse in its output.

Models based on sparse communication [64] for scalability have been proposed recently for feedforward networks, using locality sensitivity hashing to dynamically choose downstream units for communicating activations. This is a dynamic form of parameter-sparsity [25]. But, parameter/model-sparsity is, in general, orthogonal to and complementary with our method for activity-sparsity, and can easily be combined for additional gains.

Biologically realistic spiking networks [41] are often implemented using event-based updates and have been scaled to huge sizes [33], albeit without any task-related performance evaluation. Models for deep learning with recurrent spiking networks [3, 55] mostly focus on modeling biologically realistic memory and learning mechanisms. Moreover, units in a spiking neural network implement dynamics based on biology and communicate solely through unitary events, while units in an EGRU send real-valued signals to other units, and have more general dynamics. A sparse learning rule was recently proposed [4] that is a local approximation of backpropagation through time, but not event-based.

The event-based learning rule for the continuous time EGRU is inspired by, and a generalization of, the event-prop learning rule for spiking neurons [63]. As in that paper, we use the adjoint method for ordinary differential equations (ODEs) to train the continuous time EGRU [10, 50] combined with sensitivity analysis for hybrid discrete/continuous systems [11, 19]. Using pseudo-derivatives for back-propagating through the non-differential threshold function, as we use for our discrete-time EGRU, was originally proposed for feedforward spiking networks in neuromorphic hardware in [16] and developed further in [3, 65]. The sparsity of learning with BPTT when using appropriate pseudo-derivatives in a discrete-time feed-forward spiking neural network was recently described in [49].

A continuous time version of sigmoidal RNNs was first proposed in [2] and for GRUs in [14]. The latter used a Bayesian update for network states when input events were received, but the network itself was not event-based. As in [37, 46], the focus there was on modeling irregularly spaced input data, and not on event-based network simulation or activity-sparse inference and training. [9] also recently proposed a continuous time recurrent network for more stable learning, without event-based mechanics. GRUs were formulated in continuous time in [32], but purely for analyzing its autonomous dynamics.

## 3 Event-based GRU

### 3.1 Time-sparse GRU formulation

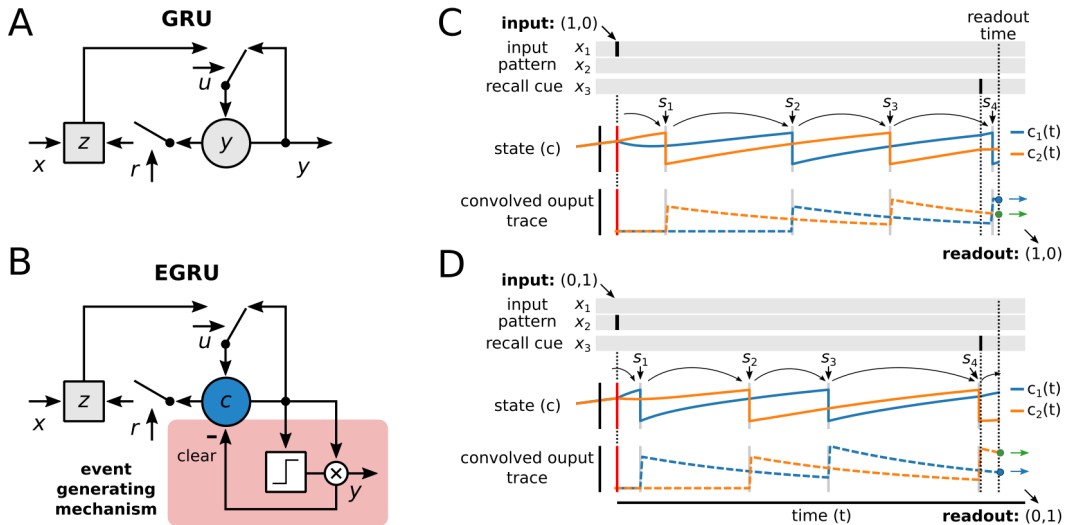

**Figure 1:** Illustration of EGRU. **A:** A single unit of the original GRU model adapted from [12]. **B:** EGRU unit with event generating mechanism. **C,D:** Dynamics of EGRU internal state variables for the delay-copy task with input (1,0) (**C**) and (0,1) (**D**). Colors are matched for neurons in both plots.

We base our model on the GRU [12], illustrated for convenience in Fig. 1A. It consists of internal gating variables for updates ($\mathbf{u}$) and a reset ($\mathbf{r}$), that determine the behavior of the internal state $\mathbf{y}$. The state variable $\mathbf{z}$ determines the interaction between external input $\mathbf{x}$ and the internal state. The dynamics of a layer of GRU units, at time step $t$, is given by the set of vector-valued update equations:

$$\mathbf{u}^{\langle t \rangle} = \sigma\Big(\mathbf{W}_u\Big[\mathbf{x}^{\langle t \rangle}, \mathbf{y}^{\langle t-1 \rangle}\Big] + \mathbf{b}_u\Big), \quad \mathbf{r}^{\langle t \rangle} = \sigma\Big(\mathbf{W}_r\Big[\mathbf{x}^{\langle t \rangle}, \mathbf{y}^{\langle t-1 \rangle}\Big] + \mathbf{b}_r\Big),$$

$$\mathbf{z}^{\langle t \rangle} = g\Big(\mathbf{W}_z\Big[\mathbf{x}^{\langle t \rangle}, \mathbf{r}^{\langle t \rangle} \odot \mathbf{y}^{\langle t-1 \rangle}\Big] + \mathbf{b}_z\Big), \quad \mathbf{y}^{\langle t \rangle} = \mathbf{u}^{\langle t \rangle} \odot \mathbf{z}^{\langle t \rangle} + (1 - \mathbf{u}^{\langle t \rangle}) \odot \mathbf{y}^{\langle t-1 \rangle},$$

(1)

117 where $\mathbf{W}_{u/r/z}$, $\mathbf{b}_{u/r/z}$ denote network weights and biases, $\odot$ denotes the element-wise (Hadamard)
118 product, and $\sigma(\cdot)$ is the vectorized sigmoid function. The notation $\left[\mathbf{x}^{\langle t\rangle},\mathbf{y}^{\langle t-1\rangle}\right]$ denotes vector concate-
119 nation. The function $g(\cdot)$ is an element-wise nonlinearity (typically the hyperbolic tangent function).

120 We introduce an event generating mechanisms by augmenting the GRU with a rectifier and a clearing
121 mechanism (see Fig. 1B for an illustration). This introduces an event-based variant of the internal
122 state variable $y_i^{\langle t\rangle}$, that is nonzero when the internal dynamics reach a threshold $\vartheta_i$ and is cleared
123 immediately afterwards. Formally, this can be included in the model by adding an auxiliary internal
124 state $c_i^{\langle t\rangle}$, and replacing $\mathbf{y}^{\langle t\rangle} = (y_1^{\langle t\rangle}, y_2^{\langle t\rangle},...)$ with the event-based form

$$y_i^{\langle t\rangle} = c_i^{\langle t\rangle} H\left(c_i^{\langle t\rangle} - \vartheta_i\right) \quad \text{with} \quad c_i^{\langle t\rangle} = u_i^{\langle t\rangle} z_i^{\langle t\rangle} + (1 - u_i^{\langle t\rangle}) c_i^{\langle t-1\rangle} - y_i^{\langle t-1\rangle}, \tag{2}$$

125 where $H(\cdot)$ is the Heaviside step function and $\vartheta_i > 0$ is a trainable threshold parameter. This form
126 is well suited for time sparsity, since $H(\cdot)$ acts here as a gating mechanism, by generating a single
127 non-zero output when $c_i^{\langle t\rangle}$ crosses the threshold $\vartheta_i$. That is, at all time steps $t$ with $c_i^{\langle t\rangle} < \vartheta_i, \forall i$, we
128 have $y_i^{\langle t\rangle} = 0$. The $-y_i^{\langle t-1\rangle}$ term in Eq. (2) makes emission of multiple consecutive events by the same
129 unit unlikely, hence favoring overall sparse activity. With this formulation, each unit only needs to
130 be updated when an input is received either externally or from another unit in the network. This is
131 because, if both $x_i^{\langle t\rangle} = y_i^{\langle t-1\rangle} = 0$ for the $i$-th unit, then $u_i^{\langle t\rangle}, r_i^{\langle t\rangle}, z_i^{\langle t\rangle}$ are essentially constants, and
132 hence the update for $y_i^{\langle t\rangle}$ can be retroactively calculated efficiently on the next incoming event.

### 3.2 Limit to continuous time

134 The discrete time model Eq. (1) considers the GRU dynamics only at integer time points,
135 $t_0 = 0, t_1 = 1, t_2 = 2,....$ However, in general it is possible to express the GRU dynamics for an arbitrary
136 time step $\Delta t$, with $t_n = t_{n-1} + \Delta t$. The discrete time GRU dynamics can be intuitively interpreted
137 as an Euler discretization of an ordinary differential equation (ODE) [32] (see Supplement), which
138 we extend further to formulate the EGRU. This is equivalent to taking the continuous time limit $\Delta t \to 0$
139 to get dynamics for the internal state $\mathbf{c}(t)$ starting from the discrete time EGRU model outlined above.
140 In the resulting dynamical system equations inputs cause changes to the states only at the event times,
141 whereas the dynamics between events can be expressed through ODEs. To arrive at the continuous
142 time formulation we introduce the neuronal activations $\mathbf{a}_u(t)$, $\mathbf{a}_r(t)$ and $\mathbf{a}_z(t)$, with

$$\mathbf{u}(t) = \sigma(\mathbf{a}_u(t)), \quad \mathbf{r}(t) = \sigma(\mathbf{a}_r(t)), \quad \mathbf{z}(t) = g(\mathbf{a}_z(t)),$$
$$\text{with dynamics} \quad \tau_s \dot{\mathbf{a}}_{\mathrm{X}} = -\mathbf{a}_{\mathrm{X}} - \mathbf{b}_{\mathrm{X}}, \quad \mathrm{X} \in \{u, r, z\} \tag{3}$$

143 and

$$\tau_m \dot{\mathbf{c}}(t) = \mathbf{u}(t) \odot (\mathbf{z}(t) - \mathbf{c}(t)) = F(t, \mathbf{a}_u, \mathbf{a}_r, \mathbf{a}_z, \mathbf{c}), \tag{4}$$

144 where $\tau_s$ and $\tau_m$ are time constants, $\mathbf{c}(t)$, $\mathbf{u}(t)$ and $\mathbf{z}(t)$ are the continuous time analogues to $\mathbf{c}^{\langle t\rangle}$,
145 $\mathbf{u}^{\langle t\rangle}$ and $\mathbf{z}^{\langle t\rangle}$, and $\dot{\mathbf{a}}_{\mathrm{X}}$ denotes the time derivative of $\mathbf{a}_{\mathrm{X}}$. The boundary conditions are defined for $t = 0$
146 as $\mathbf{a}_{\mathrm{X}}(0) = \mathbf{c}(0) = \mathbf{0}$. The function $F$ in Eq. (4) determines the behavior of the EGRU between event
147 times, i.e. when $\mathbf{x}(t) = \mathbf{0}$ and $\mathbf{y}(t) = \mathbf{0}$. Nonzero external inputs and internal events cause jumps in
148 $\mathbf{c}(t)$ and $\mathbf{a}_{\mathrm{X}}(t)$.

149 Furthermore, the formulation of the event generating mechanisms Eq.(2) introduced above can be
150 expressed in a straightforward manner in continuous time. Note that in continuous time the exact time
151 $s$ at which the internal variable $c_i(s)$ reaches the threshold ($c_i(s) = \vartheta_i$) can be determined with very
152 high precision. Therefore, the value of $c_i(s)$ and the instantaneous amplitude of $y_i(s)$ simultaneously
153 approach $\vartheta_i$ at time point $s$, so that the $-y_i$ term in Eq. (2) effectively resets $c_i(s)$ to zero, right after
154 an event was triggered. To describe these dynamics we introduce the set of internal events $\mathbf{e}$, $e_k \in \mathbf{e}$,
155 $e_k = (s_k, n_k)$, where $s_k$ are the continuous (real-valued) event times, and $n_k$ denotes which unit got
156 activated. An event $e_k$ is triggered whenever $c_{n_k}(t)$ reaches $\vartheta$. More precisely:

$$(s_k, n_k) : c_{n_k}^-(s_k) = \vartheta_{n_k}, \tag{5}$$

157 where the superscript $.^-$ ($.^+$) denotes the quantity just before (after) the event. Immediately after
158 an event has been generated the internal state is cleared: $c_{n_k}^+(s_k) = 0$. At the time of this event, the

activations of all the units $m \neq n_k$ connected to unit $n_k$ experiences a jump in its state value. The jump for $a_{X,m}$ is given by:

$$a_{X,m}^+(s_k) = a_{X,m}^-(s_k) + w_{X,mn_k} r_{X,n_k} c_{n_k}^-(s_k),$$ (6)

where $X \in \{u, r, z\}$, $\mathbf{r}_X = 0$ when $X \in \{u, z\}$ and $\mathbf{r}_X = \mathbf{r}$ when $X = \{r\}$. This is equivalent to $y_i = c_{n_k}^-$ being the output of each network unit. A similar jump is experienced on arrival of an external input, using the appropriate input weights instead (see Supplement for specifics).

The continuous time event-based state update is illustrated in Fig. 1C and D for the delay-copy task described in Section 4.1. Two EGRU units are used here and states $c_i(t)$ and event times $s_k$ are shown. At the beginning of the trial an input pattern ($x_1 = 1$, $x_2 = 0$, and $x_1 = 0$, $x_2 = 1$ in Fig. 1C and D, respectively) has to be memorized in the network and retrieved again after the recall cue ($x_3 = 1$) was given. The parameters are trained with the event-based updates described in Section 3.3. The required memory is stored in the internal events and state dynamics. State updates can be performed in an event-based fashion, i.e. by jumping from one event time $s_k$ to the next $s_{k+1}$. In-between state values follow the state dynamics Eq.(4) and their values are not needed to perform the updates (but are shown here for the sake of illustration). By construction, the state updates for external and internal events only happen on receipt of event. Since Eqs. (3), (4) are linear ODEs, the intermediate updates due to autonomous state dynamics can also be performed cumulatively and efficiently just at event times, hence avoiding any computation in the absence of incoming events.

## 3.3 Event-based gradient-descent using adjoint method

To derive the event-based gradient updates for the EGRU we define the loss over duration $T$ as $\int_0^T \ell_c(\mathbf{c}(t),t)dt$, where $\ell_c(\mathbf{c}(t),t)$ is the instantaneous loss at time $t$. $T$ is a task-specific time duration within which the training samples are given to the network as events, and the outputs are read out. In general $\ell_c(\mathbf{c}(t),t)$ may depend arbitrarily on $\mathbf{c}(t)$, however in practice we choose the instantaneous loss to depend on the EGRU states only at specific output times to adhere to our fully event-based algorithm.

The loss is augmented with the terms containing the Lagrange multipliers $\boldsymbol{\lambda}_c, \boldsymbol{\lambda}_{a_x}$ to add constraints defining the dynamics of the system from Eqs. (3), (4). The total loss $\mathcal{L}$ thus reads

$$\mathcal{L} = \int_0^T \left[ \ell_c(\mathbf{c}(t),t) + \boldsymbol{\lambda}_c \cdot (\tau_m \dot{\mathbf{c}}(t) - F(t,\mathbf{a}_u,\mathbf{a}_r,\mathbf{a}_z,\mathbf{c})) + \sum_{X \in \{u,r,z\}} \boldsymbol{\lambda}_{a_x} \cdot (\tau_s \dot{\mathbf{a}}_X + \mathbf{a}_X) \right] dt.$$ (7)

The Lagrange multipliers are referred to as the adjoint variables in this context, and may be chosen freely since both $\tau_m \dot{\mathbf{c}}(t) - F(t,\mathbf{a}_u,\mathbf{a}_r,\mathbf{a}_x,\mathbf{c})$ and $\tau_s \dot{\mathbf{a}}_X + \mathbf{a}_X$ are everywhere zero by construction.

We can choose dynamics and jumps at events for the adjoint variables in such a way that they can be used to calculate the gradient $\frac{d\mathcal{L}}{dw_{ji}}$. Calculating the partial derivatives taking into account the discontinuous jumps at event times depends on the local application of the implicit function theorem, which requires event times to be a differentiable function of the parameters. See [10, 19, 63] for a description of applying the adjoint method for hybrid discrete/continuous time systems with further theoretical background, and the Supplement for a derivation specific to the EGRU.

The time dynamics of the adjoint variables is given by the following equations with a boundary condition of $\boldsymbol{\lambda}_c(T) = \boldsymbol{\lambda}_{a_x}(T) = 0$:

$$\left(\frac{\partial F}{\partial \mathbf{c}}\right)^T \boldsymbol{\lambda}_c - \tau_m \dot{\boldsymbol{\lambda}}_c = 0, \qquad \boldsymbol{\lambda}_{a_x} + \left(\frac{\partial F}{\partial \mathbf{a}_x}\right)^T \boldsymbol{\lambda}_c - \tau_s \dot{\boldsymbol{\lambda}}_{a_x} = 0,$$ (8)

for $X \in \{u,r,z\}$, and $M^T$ denoting the transpose of the matrix $M$. The event updates for the adjoints are described in the Supplement. In practice, the integration of $\boldsymbol{\lambda}$ is done backwards in time.

For the recurrent weights $w_{X,ij}$ from the different parameter matrices $W_X$ for $X \in u,r,z$, we can write the weight updates using only quantities calculated at events $e_k$ as:

$$\Delta w_{X,ij} = \frac{\partial}{\partial w_{X,ij}} \mathcal{L}(\mathbf{W}) = \sum_k \xi_{X,ijk}.$$ (9)

The corresponding value of $\xi_{X,ijk} = (\boldsymbol{\xi}_{X,k})_{ij}$ is given by the following formula, written in vector form for succinctness:

$$\boldsymbol{\xi}_{X,k} = -\tau_s \left( \mathbf{r}_X^-(s_k) \odot \mathbf{c}^-(s_k) \right) \otimes \boldsymbol{\lambda}_{a_x}^+(s_k),$$ (10)

200  where $\otimes$ is the outer product, $\mathbf{c}^-$ refers to the value of $\mathbf{c}(t)$ just before event $e_k$, $\mathbf{r}_x^- = 0$ for $x \in \{u, z\}$
201  and equal to the value of $\mathbf{r}(t)$ just before event $e_k$ for $x = \{r\}$, $\boldsymbol{\lambda}_{a_x}^+$ refers to the value of the adjoint
202  variable $\boldsymbol{\lambda}_{a_x}(t)$ just after the event $e_k$. Thus, the values of $\mathbf{r}(t), \mathbf{c}(t)$ needs to be stored only at event
203  times, and $\boldsymbol{\lambda}_{a_x}(t)$ needs to be calculated only at these times, making the gradient updates event-based.
204  See the Supplement for the update rules for the input weights and biases.

## 3.4  Sparse approximate BPTT in discrete time

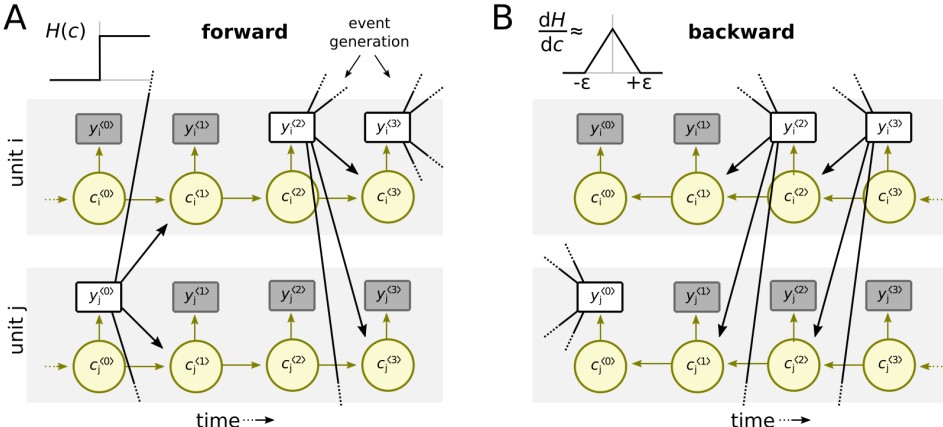

**Figure 2:** Illustrate the discrete time state dynamics for two EGRU units ($i$ and $j$). **A:** Forward dynamics. Information only propagates from units that generate an event. **B:** Activity-sparse backward dynamics. Insets show threshold function $H(c)$ and pseudo derivative thereof.

206  In discrete time, the network uses a threshold activation function $H(c)$ to decide whether to emit an event
207  as described in Eq. (2). Since $H(c)$ is not differentiable at the threshold $\vartheta_i$, we define a pseudo-derivative
208  at that point for calculating the backpropagated gradients. The pseudo-derivative is defined as a piece-
209  wise linear function that is non-zero for values of state $c_i$ between $\vartheta_i + \varepsilon$ and $\vartheta_i - \varepsilon$ as shown in the inset
210  in Fig. 2B. Since the pseudo-derivative is zero whenever the internal state of the unit is below $\vartheta_i - \varepsilon$, the
211  backpropagated gradients are also 0 for all such units, making the backward-pass sparse (see Fig.2 for
212  an illustration). Note that the case where the internal unit state is above $\vartheta_i + \varepsilon$ tends to occur less often,
213  since the unit will emit an event and the internal state will be cleared (Eq. (2)) at the next simulation step.

## 3.5  Computation and memory reduction due to sparsity

215  For the forward pass of the discrete time EGRU, an activity sparsity of $\alpha$ (i.e. an average of $\alpha$ events
216  per simulation step) leads to the reduction of multiply-accumulate operations (MAC), by factor $\alpha$.
217  We focus on MAC operations, since they are by far the most expensive compute operation in these
218  models. If optimally implemented an activity sparsity of 80% will require 80% fewer MAC operations
219  compared to a GRU that is not activity-sparse. Computation related to external input is only performed
220  at input times, and hence is as sparse as the input, both in time and space. During the backward pass, a
221  similar factor of computational reduction is observed, based on the backward-pass sparsity $\beta$ which is,
222  in general, less than $\alpha$. This is because, when the internal state value is not within $\pm\varepsilon$ of the threshold
223  $\vartheta$, the backward pass is skipped, as described in section 3.4. Since our backward pass is also sparse, we
224  expect to need to store only $\beta$ fraction of the activations for later use, hence also reducing the memory
225  usage. In all our experiments, we report activity-sparsity values calculated through simulations.

# 4  Results

## 4.1  Delay-copy task

228  To illustrate the behavior of the continuous-time EGRU model (Fig. 1C,D) we used a simple delay-copy
229  task (also called the copy memory task [24]). A binary vector was presented to the network at the input
230  time. This was followed by a delay period, after which the network was given a cue input indicating
231  that it should recall the input seen before. A small network with only two EGRU units was used here,
232  trained with the event-based learning rules described in Section 3.3. Right after the cue input, the
233  network had to report the memorized input pattern. EGRU outputs $y_i$ emitted at network event times
234  were convolved with an exponential kernel to retrieve output traces, which were then used to retrieve

the stored binary patterns based on their relative magnitudes. The kernel time constant was chosen to be significantly lower than the delay time such that the network had to retain the memory in the event dynamics. The binary cross-entropy loss was used to train this model until it reached perfect (100%) bitwise accuracy on this task. Fig. 1C,D shows the dynamics of the continuous-time model after training, as well as the output trace and events. The network has learned to generate events such that output traces reliably encode the stored input patterns. Supplemental Table S1 shows the robustness of the training for different sizes of inputs, networks, delay periods, all for multiple runs.

## 4.2 Gesture prediction

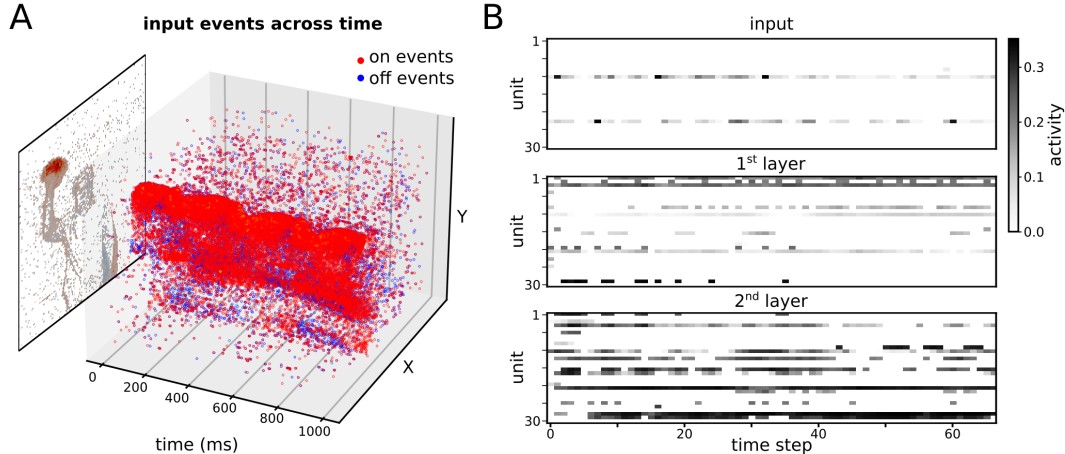

**Figure 3: A:** Illustration of DVS gesture classification data for an example class (right hand wave). On (red) and off (blue) events are shown over time and merged into a summary image for illustration (not presented to the network). **B:** Sparse activity of input and EGRU units (random subset of 30 units shown for each layer).

We next evaluate our model on gesture prediction, which is a popular real-world benchmark for RNNs. Here and in the remainder of the experiments we used the discrete time version of EGRU, since it is easier to implement and use while retaining most of the advantages of the continuous time model. We use the DVS128 Gesture Dataset [1], where the inputs are defined as events. This dataset is widely used in neuromorphic research and enables us to demonstrate our model's performance and computational efficiency on event-based data. The dataset contains 11 gestures from 29 subjects recorded with a DVS128 event camera [40]. Each event encodes a relative change of illumination and is given as spatio-temporal coordinates of X/Y position on the $128 \times 128$-pixel sensor and time stamp. Raw event times were combined into 'frames' by binning them over time windows of 25 ms. Frames were then downscaled to $32 \times 32$ pixels using a maxpool layer.

| reference | architecture (# units) | para-meters | effective MAC | accu-racy | activity sparsity | backward sparsity |
|---|---|---|---|---|---|---|
| He et al. [23] | LSTM (512) | 7.35M | 7.34M | 86.81% | - | - |
| Innocenti et al. [30] | AlexNet+LSTM+DA | 9.99M | 638.25M | 97.73% | - | - |
| **ours** | GRU (1024) | 15.75M | 15.73M | 88.07% | 0% | - |
| **ours** | **EGRU** (512) | 5.51M | 4.19M | 88.02% | 83.79% | 53.55% |
| **ours** | **EGRU** (1024) | 15.75M | 10.54M | 90.22% | 82.53% | 56.63% |
| **ours** | **EGRU**+DA (1024) | 15.75M | 10.77M | 97.13% | 78.77% | 58.20% |

**Table 1:** Model comparison for the DVS Gesture recognition task. Effective number of MAC operations as described in section 3.5.

Unlike previous approaches that focused on a feedforward/RNN hybrid approach [1, 20, 30], we focused on pure RNN based architectures following the work of [23]. A binary cross-entropy loss was applied with an additional regularization loss on the output gate to produce 5% activity and the state variable $c$ to be slightly below the threshold. The models were trained using the Adam optimizer for 1000 epochs to

verify their stability but typically reached a plateau performance after 200 epochs (see Supplement for further details). Due to this, backward sparsity as described in Section 3.5 was calculated at epoch 100.

Comparison of model performance on gesture prediction is presented in Table 1. The model had inherent activity-sparsity of 70% which the regularization increased to 90% without significant performance decrease. EGRU consistently outperformed GRU networks of the same size on this task by a small margin. Adding data augmentation (DA) by applying random crop, translation, and rotation, as previously done in [30], increased the performance to over 97% of this pure RNN architecture, coming close to state-of-the-art architectures that even include costly AlexNet pre-processing. Further experimental details, ablation studies and statistics over different runs can be found in the supplement sections D.1, D.1.1 and tables S3, S4 respectively.

### 4.3 Sequential MNIST

Next, we tested the EGRU on the sequential MNIST task [36], which is a widely used benchmark for recurrent networks. In this task, the MNIST handwritten digits were given as input one pixel at a time, and at the end of the input sequence, the network output was used to classify the digit. We trained a 1-layer EGRU with 590 units (matching the number of parameters with a 512 unit LSTM). We did not use any regularisation to increase sparsity in this task, so that we could test how much sparsity, both forward and backward, arises naturally in the EGRU. In Table 2, we report the results of discrete-time EGRU along with other state-of-the-art architectures. EGRU achieved a task performance comparable to previous architectures while using much fewer operations (more than an 5-fold reduction in effective MAC operations compared to GRU). Further experimental details, and statistics over different runs can be found in the supplement sections D.2 and table S5 respectively.

| reference | architecture (# units) | parameters | effective MAC | test accuracy | activity sparsity |
|---|---|---|---|---|---|
| Rusch and Mishra [54] | coRNN (256) | 134K | 262K | 99.4% | - |
| Gu et al. [22] | LSTM (512) | 1M | 1M | 98.8% | - |
| **ours** | GRU (590) | 1M | 1M | 98.8% | - |
| **ours** | **EGRU** (590) | 1M | 226K | 98.3% | 72.1% |

**Table 2:** Model comparison on sequential MNIST task. Top-1 test scores, given as percentage accuracy, where higher is better.

### 4.4 Language Modeling

Natural language processing is a popular domain for benchmarking recurrent neural networks. We evaluated our model on a language modeling task based on the PennTreebank [42] dataset to validate the functionality of our model. While techniques such as neural cache models [21] or dynamic evaluation [35] have been shown to improve language models, we focused on the RNN model itself in this work, taking [45] as our baseline. Following [45], our models consists of a dense 400-dimensional embedding layer, and three stacked RNN cells with DropConnect applied to the hidden-to-hidden weights [61]. The weights of the final softmax layer were tied to the embedding layer [29, 51]. All our models are optimized with Adam for 1000 epochs, and parameters were tuned for each model individually. Details on training and model parameters can be found in the Supplement. Results are shown in Table 3. In our experiments, GRUs did not reach the performance of LSTM variants on this task, which, to the best of our knowledge, is consistent with recent RNN language modeling literature [44, 45]. At the same time, EGRU slightly outperformed GRU, while maintaining high levels of activity sparsity. Further experimental details, and statistics over different runs can be found in the supplement sections D.3 and table S6 respectively.

## 5 Discussion

We have introduced EGRU, a new form of a recurrent neural network that is competitive with current deep recurrent models yet can efficiently perform both inference and learning. To achieve this, we first formulated the GRU in continuous time and converted it to an event-based form that achieved activity-sparsity naturally. Furthermore, the gradient-descent updates on this time-continuous EGRU mirrored the activity sparsity of the inference. We then demonstrated a discrete-time simplification of this model that also exhibited event-based activity-sparse inference and learning while being easier to implement with popular ML frameworks such as PyTorch or Tensorflow.

| reference | architecture (# units) | para-meters | effective MAC * | validation | test | activity sparsity |
|---|---|---|---|---|---|---|
| Gal et al. [18] | Variational LSTM | 24M | - | 77.3 | 75.0 | - |
| Melis et al. [44] | 1 layer LSTM | 24M | - | 61.8 | 59.6 | - |
| Merity et al. [45] | AWD-LSTM | 24M | 24M | 60.0 | 57.3 | - |
| **ours** | GRU (1350) | 24M | 24M | 71.2 | 68.8 | - |
| **ours** | **EGRU** (1350) | 24M | 4.7M | 67.4 | 64.5 | 88.0% |
| **ours** | **EGRU** (2000) | 45M | 6.6M | 66.5 | 63.7 | 90.4% |
| **ours** | **EGRU** (2700) | 77M | 8.1M | 66.4 | 63.5 | 93.2% |

**Table 3:** Model comparison on PennTreebank. Validation and test scores are given as perplexities, where lower is better. Sparsity refers to activity-sparsity of the EGRU output, and effective MAC operations consider the layer-wise sparsity in the forward pass. *

The EGRU achieved competitive task performance on various real-world tasks such as gesture recognition and language modeling while achieving a sparsity of up to 80% for the gesture recognition task and 90% on the language modeling task. Scaling up networks for language modeling has shown some of the most promising results in the last few years [8, 17] Hence our choice of task, albeit on a smaller scale, was to validate the functionality of the model. Considering the need for extensive hyperparameter search [44] for language modeling, our model achieved promising results while maintaining a high degree of activity-sparsity. For example, our EGRU with 1350 hidden units reached perplexities comparable with LSTM and GRU, while maintaining an activity-sparsity of $86\%$ (14% of the units active on average). The amount of computation used by an EGRU also scales sub-linearly with an increase in the size of the network and number of parameters, making it a scalable alternative to LSTM/GRU based architectures (see Supplement).

While we use the GRU as the basis for our model due to its simplicity, this formulation can easily be extended to any arbitrary network dynamics, including the LSTM, allowing specialized architectures for different domains. The adjoint method for hybrid systems that we use here is a powerful general-purpose tool for training event-based activity-sparse forms of various recurrent neural network architectures. Another novel outcome of this paper is that this theory can handle inputs in continuous time as events, which is very intuitive, hence providing an alternative to the more complex controlled differential equations [34]. The EGRU can also be used for irregularly spaced sequential data quite naturally.

The compute efficiency of this model can directly translate into gains in energy efficiency when implemented using event-based software primitives. These same properties would also allow the model to work well on heterogenous compute resources, including pure CPU nodes, and neuromorphic devices such as Intel's Loihi [13] and SpiNNaker 2 [27], that can achieve orders of magnitude higher energy efficiency. The EGRU model will also perform well in more mainstream deep learning hardware that is enabled for dynamic sparsity, such as the Graphcore system [31]. On neuromorphic devices with on-chip memory in the form of a crossbar array, the activity sparsity directly translates into energy efficiency. For larger models that need off-chip memory, activity-sparsity needs to be combined with parameter-sparsity to reduce energy-intensive memory access operations.

In summary, starting with the motivation of building scalable, energy-efficient deep recurrent models, we demonstrated the EGRU, which reduces the required compute for both inference and learning by enhancing sparsity in the network. This approach lays the foundation for exploring novel capabilities that can emerge from scaling up RNNs similar to what has been seen for feed-forward architectures in recent years.

**Potential negative societal impact:** The proposed model is a new variant of the previously published GRU and would therefore essentially inherit all potential negative societal impacts from that model, including the potential risks that come with automated surveillance systems, vulnerability to fraud and adversarial attacks, etc. (see [6] and [62] for critical reviews). However, the model also provides the potential societal benefit of making these models more energy-efficient and thus reducing the energy and carbon footprint of machine learning. Scaling this model to larger sizes, especially for language modeling, can lead to the same problems as current large language models [5]. The effect of activity sparsity on prediction bias needs to be studied further in the same way as for parameter sparsity [26].

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
