# OpenReview forum: "EGRU: Event-based GRU for activity-sparse inference and learning"
_NeurIPS.cc/2022/Conference — NeurIPS 2022 Submitted_

### Official Review · Reviewer_egpJ · 2022-07-10

**Rating:** 5
**Confidence:** 4
**Soundness:** 3 good
**Presentation:** 3 good
**Contribution:** 3 good

**Summary:**

This paper proposes an event-based continuous-time variant of the GRU model, named EGRU, motivated by building scalable, energy-efficient deep recurrent models. The EGRU unit only outputs events when the internal states reach a trainable threshold, which makes it activity-sparsity naturally. Furthermore, the authors derive corresponding BP algorithms for updating EGRU and extend it to a discrete-time version that is easier to implement by popular machine learning architecture. As the experiments show, the EGRU model exhibits comparable performance to state-of-the-art recurrent network architectures for several tasks.

**Questions:**

This paper presents a novel GRU architecture, which balances the performance and computational complexity.
- However, in the downstream application with EGRU, some experiments’ setups are unclear, as listed in Weakness. I expect the authors to clarify these points.
- Besides, the authors apply EGRU in some special cases. Since the LSTM/GRU are applied widely in CV/NLP applications, can they be replaced by EGRU to reduce the computation complexity in general?
- As GRU is simplified from LSTM, can event generating mechanism transfer to the LSTM model?


**Limitations:**

The authors have addressed the limitations and potential negative societal impact of their work in discussion section.

**Strengths And Weaknesses:**

This paper is well-written, consistent, and produces good results. By introducing the event generating mechanism, the EGRU model not only significantly reduces the multiply-accumulate operations but also preserves the performance that is comparable to the more computation-expensive GRU-based model.

Strengths:
+ The event generating mechanism is a novel module inserted into the GRU model, which is proved to effectively reduce the computation of the original model.
+ The authors conduct three downstream applications to validate the effectiveness of the EGRU model, which are convincing and with comparable performances to previous methods.

Weakness:
- For the delay-copy task, the authors only show a simple example. How about increasing the number of EGRU units? Besides, did the events generating mechanism improve the memory ability, compared to the GRU model?
- In gesture prediction, the authors down sample frames into 32x32, which may affect the performance. I wonder how the input size affects the prediction accuracy with EGRU model. In general, the larger-resolution images contain more detailed information, which could be helpful for classification.
- For language modeling experiments, the performance decreases with the increment of EGRU units. Can the authors give some explanation or the trend of performance related to the number of EGRU units?

---

> ### Author Response · Authors · 2022-08-02
> **Response to Reviewer egpJ**
>
> The authors would like to thank the reviewer for recognizing the novely of our approach and the robustness of our empirical results.
>
> > For the delay-copy task, the authors only show a simple example. How about increasing the number of EGRU units? Besides, did the events generating mechanism improve the memory ability, compared to the GRU model?
>
> The simple version of the delay-copy task was chosen for the sake of illustration (Fig.1C,D). EGRU of course also works on more complex tasks. To demonstrate this we have added additional experiments for the delay-copy task with different number of EGRU units in the supplement.
>
> The EGRU has comparable memory ability compared to the GRU model. We have added further experiments comparing the discrete time EGRU with the GRU on memory ability in Table S2 of the supplement to show this.
>
> > In gesture prediction, the authors down sample frames into 32x32, which may affect the performance. I wonder how the input size affects the prediction accuracy with EGRU model. In general, the larger-resolution images contain more detailed information, which could be helpful for classification.
>
> The down-sampling was done here to get a fair comparison with previous work that used the same preprocessing. We now have added additional experiments in Table S3 for input-size 128x128. The performance drops slightly (by 2 %) with larger input size due to the noisy nature of the dataset.
>
> > For language modeling experiments, the performance decreases with the increment of EGRU units. Can the authors give some explanation or the trend of performance related to the number of EGRU units?
>
> The observation of decreasing model performance with model size is due to overfitting. We resolved this issue in the current version of the paper. Our experiments demonstrate that the support of the pseudo-derivative (epsilon) acts as a regularizing parameter (see Figure S5). Overfitting is a known issue in language modeling, and regularization is the major component in successful LSTM language models (Merity et al, Melis et al)
>
> > Besides, the authors apply EGRU in some special cases. Since the LSTM/GRU are applied widely in CV/NLP applications, can they be replaced by EGRU to reduce the computation complexity in general?
>
> In our paper, we test the EGRU on gesture recognition using data from dynamic vision sensors, sequential MNIST and Penn Treebank, all of which broadly fall in the domain of CV and NLP. Therefore, we do expect that we can replace LSTM/GRU with EGRU in these domains to reduce the computational complexity.
>
> > As GRU is simplified from LSTM, can event generating mechanism transfer to the LSTM model?
>
> Yes, the event generating mechanism can be transferred to the LSTM quite easily by adding the thresholding function after the output gate. We haven’t performed any experiments on this setup yet, but plan to do so in the future.

---

### Official Review · Reviewer_RdzD · 2022-07-11

**Rating:** 6
**Confidence:** 2
**Soundness:** 3 good
**Presentation:** 3 good
**Contribution:** 3 good

**Summary:**

This paper proposes EGRU, an event-based GRU for activity-sparse inference and learning. The proposed model is more efficient as only a select set of units are active at any given time.

**Questions:**

* In terms of activity sparsity during inference, how does it compare against pruning a GRU, such as the one mentioned in "EXPLORING SPARSITY IN RECURRENT NEURAL NETWORKS"?

**Limitations:**

The authors have discussed inheriting negative societal impacts and risks existed in GRU.

**Strengths And Weaknesses:**

**Originality**:

* To my knowledge, the idea proposed in this article is novel and original.

**Quality**:

* The paper technically sounds correct and claims well supported by theoretical analysis and experimental results.
* Related works are covered and discussed.
* Experiments are conducted extensively in different type of tasks and the results are discussed thoroughly and compared in terms of accuracy,#MAC, and sparsity.


**Clarity**:

* This paper is well written and organised.


**Significance**:
* the proposed EGRU, as the authors have suggested, can be more energy-efficient and scalable as it reduces the required compute for both inference and learning.

---

> ### Author Response · Authors · 2022-08-02
> **Response to Reviewer RdzD**
>
> The authors would like to thank the reviewer for the positive review and recognising that the paper is well written and the approach novel.
>
> > In terms of activity sparsity during inference, how does it compare against pruning a GRU, such as the one mentioned in "EXPLORING SPARSITY IN RECURRENT NEURAL NETWORKS"?
>
> The authors would like to thank the reviewer for pointing out this reference. This reference primarily deals with parameter sparsity rather than activity sparsity that is the topic of the present paper. But the authors agree with the reviewer that this paper should be discussed in the related work, and we plan to do so in our final version.
>
> The suggested reference implements a weight magnitude pruning method and apply it on the GRU, where they report 88.6% sparsity resulting in a 13.8% drop in performance for a GRU with 2560 units. We train smaller models, and see a smaller drop in performance with high levels of activity sparsity.
>
> The authors would like to stress that activity and parameter sparsity are orthogonal and could be combined in future work, which would likely lead to further reducing the required resources above the level each individual method could achieve (although we do not combine them in our paper to avoid enlarging the scope of the paper too much).

---

### Official Review · Reviewer_eSbV · 2022-07-12

**Rating:** 6
**Confidence:** 4
**Soundness:** 3 good
**Presentation:** 3 good
**Contribution:** 3 good

**Summary:**

In this paper, the authors have introduced an event-based continuous-time variant of the gated recurrent model and derived an eve-based form of backprop to train the system. Extensive experiments on a wide variety of benchmarks show that the proposed model EGRU can achieve comparable performance with SOTA recurrent models and exhibits high levels of activity-sparsity during the training and inference phase.

**Questions:**

* The comparison should be made with at least two models from [1-5], first, one proposed sparse RTRL to train neural networks, that can potentially solve the scalability issues shown by BPTT.  Other [2] focuses on other recurrent architecture and learning approaches to solve this issue.

* For models only based on BPTT, prior work has shown that sparse versions of recurrent models have the potential to outperform dense versions. Hence it is important to show how efficient the current approach is compared to other sparse RNNs.

*Hyper-parameter optimization- No detail about hyper-parameter optimization, authors should provide reasoning how these parameters were chosen.

* Statistical significance: - Experiments should be conducted for K trials and average performance and the standard error should be reported.
* Can authors comment on the robustness and stability of the given model? Based on different seeds, and settings, the optimizer is model stable compared to dense models trained using BPTT.

* Authors have mentioned that EGRU is computationally efficient, can authors provide numbers in FLOPS for both training and inference compared with other sparse and dense RNNs

[1] https://arxiv.org/abs/2006.07232
[2] https://arxiv.org/abs/1602.03032
[3] https://arxiv.org/pdf/1901.09208.pdf
[4] https://link.springer.com/article/10.1007/s00521-021-05727-y
[5] https://proceedings.mlr.press/v139/liu21p.html


**Limitations:**

Highlighted above

**Strengths And Weaknesses:**

# Strengths
* Approach is novel
* Well-written paper
* Experiments are good

# Weakness
* Ablation study missing
* Experimental section is missing key details
* Missing some key citations

---

> ### Author Response · Authors · 2022-08-02
> **Response to Reviewer eSbV (part 1/2)**
>
> The authors would like to thank the reviewer for the valuable comments and recognizing that the paper is well written and the approach novel.
>
> Addressing the questions of the reviewer:
>
> > Statistical significance: - Experiments should be conducted for K trials and average performance and the standard error should be reported.
>
> > Can authors comment on the robustness and stability of the given model? Based on different seeds, and settings, the optimizer is model stable compared to dense models trained using BPTT.
>
> We would like to point the reviewer to the supplement which already contained statistics over multiple independent random seeds and various different hyper-parameters in the submitted version of the paper. We have added additional studies that demonstrates that the model is robust against modifications and parameter variations — see tables S3, S5, S6 and figure S5.
>
> An ablation study of the model features also *had already been included* in the submitted version of the paper (see supplement section D.1.1 and Table S4). In the updated version, we have further extended the ablation study (see Table S4).
>
> The supplementary text also included all details of the experiments and training procedure.
>
> We have also added more pointers to these additional experiments in the main text to make them more visible.
>
> > The comparison should be made with at least two models from [1-5], first, one proposed sparse RTRL to train neural networks, that can potentially solve the scalability issues shown by BPTT. Other [2] focuses on other recurrent architecture and learning approaches to solve this issue.
>
> The authors would like to thank the reviewer for pointing out these missing references. Although all these references concern parameter sparsity rather than activity sparsity that is the topic of the present paper, the authors agree that a fair comparison with state of the art should include these prior methods. We plan to include these papers in the related work of our final version.
>
> The evaluation on Penn Treebank that were done in [4] and [5] are comparable to our study (Table 3), demonstrating similar performance (our best test perplexity of 63.5) and high levels of activity sparsity. The authors would like to stress that activity and parameter sparsity are orthogonal and could be combined in future work, which would likely lead to further savings of resources above the level each individual method could achieve (although we do not combine them in our paper to avoid enlarging the scope of the paper too much).
>
> In comparison to our proposed model, paper [1] combines truncated forward-mode differentiation based learning with parameter sparsity, showing that they are able to get an advantage for small truncation lengths, which are approximations to RTRL. With larger truncation lengths, or when using exact gradient descent with RTRL without truncation, their method does not achieve any sparsity during learning. Whereas, the activity-sparsity in our method is independent of the truncation length of BPTT, and only depends on the number of events in the network.
>
> Paper [2] proposes a modification to LSTM to improve its memory capabilities, but doesn’t directly deal with activity or parameter sparsity. Our work does not focus on memory capacity, and the EGRU has about the same memory capacity as the GRU on which it’s based, as our experimental results suggest.
>
> Paper [3] proposes a form of parameter sparsity in LSTMs, and are able to train the network with parameter sparsity throughout the whole training process. In our case, we do have activity-sparsity throughout the training process.
>
> Paper [4] proposes a method for parameter sparsity in a general RNN and the ability to keep a fixed FLOP budget using pruning and regrow in every iteration. The results reported for LSTMs in [4] can be closely compared to our results. The EGRU achieves a better model perplexity on the Penn Tree Bank (see Table 3 and Table S6) at high levels of activity sparsity. The level of sparsity is similar to the reduction in FLOPs that would be expected for EGRU, but the EGRU doesn’t provide the ability to maintain a fixed FLOP budget (fluctuations around some mean value due to the number of events are expected).
>
> Paper [5]  introduces a sparse-to-sparse training algorithm that is build around parameter sparsity.  In their case, the best stacked LSTM approach achieves 73.5 perplexity on PTB at a sparsity of 62%, whereas the best EGRU model achieves 63.5 test perplexity with high activity sparsity.
>
> We plan to add references and discussions about these related works to the final paper.

---

> > ### Author Response · Authors · 2022-08-02
> > **Response to Reviewer eSbV (part 2/2)**
> >
> > > Hyper-parameter optimization- No detail about hyper-parameter optimization, authors should provide reasoning how these parameters were chosen.
> >
> > We have now added additional information about our hyper-parameter optimization to the supplement
> >
> > > Authors have mentioned that EGRU is computationally efficient, can authors provide numbers in FLOPS for both training and inference compared with other sparse and dense RNNs
> >
> > The authors would like to point out that the initial submission included an analysis in terms of reduction in MAC operations, which is a commonly used performance indicator for algorithms run on GPUs. Tables 1-3 include effective MAC operations combined with reported activity sparsity levels. There is a close relationship between MACs and FLOPs, as in the most general case, 1 MAC = 2 FLOPs. On some platforms that support the Fused Multiply Add (FMA) operation, 1 MAC = 1 FLOP. In both cases, our MAC can be directly converted to FLOPs for comparison. The only additional operation that we introduce on top of a GRU is the thresholding function, which uses k FLOPs (comparisons) at every timestep.
> >
> > For training, BPTT uses $Tk+p, k^2+p$ for memory and compute, whereas SnAp-1 is $k+dp, d(k^2+p)$ and SnAp-2 is $k+d^2kp,d(k^2+d^2k^2p)$ resp. where T is length of sequence, k is no. of neurons, p is no. of parameters, d is density of parameters (sparsity is $1-d$).
> >
> > In the case of continuous time EGRU, all operations are further multiplied by the event-sparsity $\alpha$, with everything else remaining the same. In this sense, parameter and activity-sparsity are composable.

---

> > > ### Comment · Reviewer_eSbV · 2022-08-09
> > > **Thank you for your response**
> > >
> > > I have raised my score and the authors have done good work in the rebuttal.

---

### Author Response · Authors · 2022-08-02
**Summary of updates**

We would like to thank all their reviewers for their constructive comments and questions. Please note that we have uploaded an updated version of our main text as well as supplement. We have indicated all major changes using blue color text. We have also responded to each reviewer separately and in detail.

The updates are summarized as follows:

1. We have added pointers to the ablation studies and statistics over multiple runs in the main text.
2. We have added additional runs in the supplement for continuous time EGRU with different settings (Table S1), and comparing the memory ability of discrete time EGRU with GRU (as requested by egpJ).
3. We have added additional experiments to the DVS and PTB tasks in the supplement, to further demonstrate the performance and robustness of the EGRU.
4. We will add the references suggested by reviewer eSbV and RdzD in the final version (due to space constraints we are unable to add them right now).
5. We have added details about how we performed the hyper-parameter search for our experiments as requested by reviewer eSbV in the supplement.

---

### Meta-Review · Area_Chair_35a6 · 2022-08-25

**Recommendation:** Reject
**Confidence:** Certain

**Metareview:**

This paper introduces an event-based GRU to obtain an efficient continuous-time RNN. Although the method is sound and can work on a series of small sequence modeling tasks, there are multiple issues with the significance of the results of the paper which I point out in the following:

1) There have been significant advances in recurrent neural networks designed to efficiently model sequences, and their scaling properties, which are overlooked in this paper. In particular, structural state-space models [1], diagonal state-space models [2], LSSL [3], Closed-form continuous-time networks [4], efficient memorization via polynomial projection [5], and Neural Rough DEs [6] are currently shaping the state-of-the-art sequence modeling frameworks that efficiently model tasks with long-range dependencies while significantly outperforming Transformers and their variants. Therefore, from the perspective of representation learning capabilities, there is a significant gap between what EGRU (proposed in this paper) could achieve compared to the state-of-the-art sequence modeling tools powered by recurrent networks. Before getting published, it is essential to compare performance and speed to these models on proper benchmarks such as Long Range Arena [7].

[1] Gu, A., Goel, K., & Ré, C. (2021). Efficiently modeling long sequences with structured state spaces. arXiv preprint arXiv:2111.00396

[2] Gupta, A. (2022). Diagonal State Spaces are as Effective as Structured State Spaces. arXiv preprint arXiv:2203.14343.

[3] Gu, A., Johnson, I., Goel, K., Saab, K., Dao, T., Rudra, A., & Ré, C. (2021). Combining recurrent, convolutional, and continuous-time
models with linear state space layers. Advances in neural information processing systems, 34, 572-585.

[4] Hasani, R., Lechner, M., Amini, A., Liebenwein, L., Tschaikowski, M., Teschl, G., & Rus, D. (2021). Closed-form continuous-depth models. arXiv preprint arXiv:2106.13898.

[5] Gu, A., Dao, T., Ermon, S., Rudra, A., & Ré, C. (2020). Hippo: Recurrent memory with optimal polynomial projections. Advances in Neural Information Processing Systems, 33, 1474-1487.

[6] Morrill, J., Salvi, C., Kidger, P., & Foster, J. (2021, July). Neural rough differential equations for long time series. In International Conference on Machine Learning (pp. 7829-7838). PMLR.

[7] Tay, Y., Dehghani, M., Abnar, S., Shen, Y., Bahri, D., Pham, P., ... & Metzler, D. (2020, September). Long Range Arena: A Benchmark for Efficient Transformers. In International Conference on Learning Representations.

2) When it comes to the efficiency of computations on spatiotemporal tasks, especially when using a benchmark such as DVS Gesture detection, efficient models such as spiking networks must be accounted for. For instance, in [8], authors outperform EGRU on the DVS task with over 10x fewer parameters. In this case, EGRU+DA achieves 97% accuracy with 15.75M  (10.77M MAC) parameters, while the method proposed in [8] achieves 98% accuracy with 1.1M parameters. I believe it is essential to compare results with appropriate efficient models both in terms of computational efficiency and performance.

[8] She, X., Dash, S., Mukhopadhyay, S.: Sequence approximation using feedforward spiking neural network for spatiotemporal learning: Theory and optimization methods. In: International Conference on Learning Representations (2022), https://openreview.net/forum?id=bp-LJ4y_XC

3) Selected benchmarks for testing EGRU are not appropriate. For instance, sMNIST is a solved problem already with 99% accuracy of CoRNN that even outperforms EGRU (We need more clarifications on this as well). Even DVS is almost solved (98% from [8]). Instead, I suggest the authors try to use more challenging and up-to-date benchmarks such as Long Range Arena [7], audio datasets, and larger language benchmarks.


For the above fundamental reasons, I vote for the rejection of this paper and encourage the authors to incorporate these critical points in their next submission.




**Award:**

No

---

### Decision · Program_Chairs · 2022-09-14

Reject